# Efficient Federated Random Subnetwork Training

**Berivan Isik**[*]
Department of Electrical Engineering
Stanford University
berivan.isik@stanford.edu

**Francesco Pase**[*†]
Department of Electrical Engineering
University of Padova
pasefrance@dei.unipd.it

**Deniz Gunduz**
Department of Electrical Engineering
Imperial College London
d.gunduz@imperial.ac.uk

**Tsachy Weissman**
Department of Electrical Engineering
Stanford University
tsachy@stanford.edu

**Michele Zorzi**
Department of Electrical Engineering
University of Padova
zorzi@dei.unipd.it

## Abstract

One main challenge in federated learning is the large communication cost of exchanging weight updates from clients to the server at each round. While prior work has made great progress in compressing the weight updates through gradient compression methods, we propose a radically different approach that does not update the weights at all. Instead, our method freezes the weights at their initial *random* values and learns how to sparsify the random network for the best performance. To this end, the clients collaborate in training a *stochastic* binary mask to find the optimal sparse random network within the original one. At the end of the training, the final model is a sparse network with random weights – or a subnetwork inside the dense random network. We show improvements in accuracy, communication (less than 1 bit per parameter (bpp)), convergence speed, and final model size (less than 1 bpp) over relevant baselines on MNIST, EMNIST, CIFAR-10, and CIFAR-100 datasets, in the low bitrate regime under various system configurations. (See (Isik et al., 2022a) for the full version.)

## 1 Introduction

Federated learning (FL) is a distributed learning framework where clients collaboratively train a model by performing local training on their data and by sharing their local updates with a server every few iterations, which in turn aggregates the local updates to create a global model, that is then transmitted to the clients for the next round of training. While being an appealing approach for enabling model training without the need to collect client data at the server, communication of local updates is a significant bottleneck in FL (Kairouz et al., 2021). This has motivated research in communication-efficient FL strategies (McMahan et al., 2017a) and various gradient compression schemes via sparsification (Lin et al., 2018; Wang et al., 2018; Barnes et al., 2020; Ozfatura et al., 2021; Isik et al., 2022b), quantization (Alistarh et al., 2017; Wen et al., 2017; Bernstein et al., 2018;

---

[*]First two authors contributed equally to this work.

[†]Work done while the author was a visiting researcher at the Imperial College London.

Workshop on Federated Learning: Recent Advances and New Challenges, in Conjunction with NeurIPS 2022 (FL-NeurIPS'22). This workshop does not have official proceedings and this paper is non-archival.

Mitchell et al., 2022), and low-rank approximation (Konečnỳ et al., 2016; Vargaftik et al., 2021, 2022; Basat et al., 2022).

In this work, while aiming for communication efficiency in FL, we take a radically different approach from prior work, and propose a strategy that does not require communication of weight updates. To be more precise, instead of training the weights,

1. the server initializes a dense random network with $d$ weights, denoted by the weight vector $\boldsymbol{w}^{\text{init}} = (w_1^{\text{init}}, w_2^{\text{init}}, \ldots, w_d^{\text{init}})$, using a random seed SEED, and broadcasts SEED to the clients enabling them to reproduce the same $\boldsymbol{w}^{\text{init}}$ locally,

2. both the server and the clients keep the weights frozen at their initial values $\boldsymbol{w}^{\text{init}}$ at all times,

3. clients collaboratively train a *probability mask* of $d$ parameters $\boldsymbol{\theta} = (\theta_1, \theta_2, \ldots, \theta_d) \in [0, 1]^d$,

4. the server samples a binary mask from the trained probability mask and generates a sparse network with random weights – or a subnetwork inside the initial dense random network as follows

$$\boldsymbol{w}^{\text{final}} = \text{Bern}(\boldsymbol{\theta}) \odot \boldsymbol{w}^{\text{init}}, \tag{1}$$

where $\text{Bern}(\cdot)$ is the Bernoulli sampling operation and $\odot$ the element-wise multiplication.

We call the proposed framework Federated Probabilistic Mask Training (FedPM) and summarize it in Figure 1. At first glance, it may seem surprising that there exist subnetworks inside randomly initialized networks that could perform well without ever modifying the weight values. This phenomenon has been explored to some extent in prior work (Zhou et al., 2019; Ramanujan et al., 2020; Pensia et al., 2020; Diffenderfer & Kailkhura, 2020; Aladago & Torresani, 2021) with different strategies for finding the subnetworks. However, how to find these subnetworks in a FL setting has not attracted much attention so far. Some exceptions to this are works by Li et al. (2021); Vallapuram et al. (2022); Mozaffari et al. (2021), which provide improvements in other FL challenges, such as personalization and poisoning attacks, while not being competitive with existing (dense) compression methods such as QSGD (Alistarh et al., 2017), DRIVE (Vargaftik et al., 2021), and SignSGD (Bernstein et al., 2018) in terms of *accuracy* under the same communication budget. In this work, we propose a *stochastic* way of finding such subnetworks while reaching higher accuracy at a reduced communication cost – less than 1 bit per parameter (bpp).

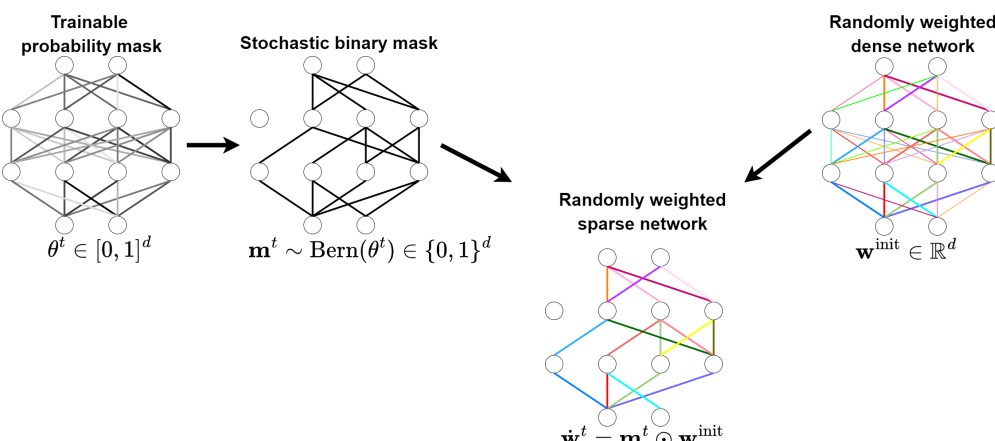

Figure 1: High-level description of extracting a randomly weighted sparse network from the trainable probability mask $\boldsymbol{\theta}^t$ in the forward-pass of round $t$ (for clients and the server). In practice, clients collaboratively train continuous scores $\boldsymbol{s} \in \mathbb{R}^d$, and then at inference time, the clients (or the server) find $\boldsymbol{\theta}^t = \text{Sigmoid}(\boldsymbol{s}^t) \in [0, 1]^d$. We skip this step in the figure for the sake of simplicity.

In addition to the accuracy and communication gains, our framework also provides an efficient representation of the final model post-training by requiring less than 1 bpp to represent (i) the random seed that generates the initial weights $\boldsymbol{w}^{\text{init}}$, and (ii) a sampled binary vector $\text{Bern}(\boldsymbol{\theta})$ (computed with the trained $\boldsymbol{\theta}$). Therefore, the final model enjoys a memory-efficient deployment – a crucial

feature for machine learning at power-constrained edge devices. Another advantage our framework brings is the privacy amplification under some settings, thanks to the stochastic nature of our training strategy. We show that in the presence of a differential privacy mechanism, such as (Abadi et al., 2016; Agarwal et al., 2021; Andrew et al., 2021), over the probability mask $\boldsymbol{\theta}$, the Bernoulli sampling step amplifies the privacy further under certain conditions.

Our contributions can be summarized as follows:

- We propose a FL framework, in which the clients do not train the model weights, but instead a stochastic binary mask to be used in sparsifying the dense network with random weights. This differs from the standard training approaches in the literature.

- Our framework provides efficient communication from clients to the server by requiring (less than) 1 bpp per client while yielding faster convergence and higher accuracy than the baselines.

- We propose a Bayesian aggregation strategy at the server side to better deal with partial participation and non-IID data splits.

- The final model (a sparse network with random weights) can be efficiently represented with a random seed and a binary mask which requires (less than) 1 bpp – at least $32\times$ more efficient storage and communication of the final model with respect to standard FL strategies.

- We demonstrate the efficacy of our strategy on MNIST, EMNSIT, CIFAR-10, and CIFAR-100 datasets under both IID and non-IID data splits; and show improvements in accuracy, bitrate, convergence speed, and final model size over relevant baselines, under various system configurations.

## 2 Federated Probabilistic Mask Training (`FedPM`)

We first describe the simpler version of the FedPM framework in Section 2.1, which provides an *unbiased* estimation of the mean of the learned probability masks at the server with *bounded error*. Next, we propose a modification in our aggregation strategy by exploiting the underlying Bernoulli mechanism in Section 2.2. This helps boost the performance of FedPM in the cases of non-IID data splits and partial participation of clients. We then discuss the details of the distribution of the initial weights in Section 2.3, and finally describe the privacy benefits of FedPM in Section 2.4. Throughout the paper, we use capital letters for random variables, small letters for their realization and deterministic quantities, and bold letters to denote vectors. Moreover, we indicate with $\boldsymbol{x}^{u,t}$ the state of the local vector $\boldsymbol{x}$ (e.g., the local mask) at client $u$ during round $t$, and with $x_i^{u,t}$ its $i$-th component. Global values are denoted with $\boldsymbol{x}^{g,t}$ and $x_i^{g,t}$, and sets are indicated with calligraphic fonts. We denote a neural network with weight vector $\boldsymbol{p}$ as $f_{\boldsymbol{p}}$.

### 2.1 `FedPM`

In this section, we present the general FedPM training pipeline. First, the server randomly initializes a neural network $f_{\boldsymbol{w}^{\text{init}}}$, parameterized by the weight vector $\boldsymbol{w}^{\text{init}} = (w_1^{\text{init}}, w_2^{\text{init}}, \ldots, w_d^{\text{init}}) \in \mathbb{R}^d$, whose components are sampled IID according to a distribution $P_{\boldsymbol{w}}$ using a *randomly generated* seed SEED. The random SEED value is then communicated to all the clients, which can locally sample the same pseudo-random vector $\boldsymbol{w}^{\text{init}}$, which is kept fixed and never modified during training. The goal for the clients is to collaboratively train a probability mask $\boldsymbol{\theta} \in [0,1]^d$, which indicates the Bernoulli parameters for the global stochastic binary mask $\boldsymbol{M} \sim \text{Bern}(\boldsymbol{\theta}) \in \{0,1\}^d$, such that the function $f_{\dot{\boldsymbol{W}}}$ maximizes its performance on a given task, where $\dot{\boldsymbol{W}} = \boldsymbol{M} \odot \boldsymbol{w}^{\text{init}}$. Specifically, FedPM learns the probabilities for the weights of being active, which are given by the probability mask $\boldsymbol{\theta} = (\theta_1, \theta_2, \ldots, \theta_d) \in [0,1]^d$. To achieve this, at every round $t$, the server samples a set $\mathcal{K}_t$ of $|\mathcal{K}_t| = K$ participants (out of the total $N$ clients), which individually train their local probability masks $\boldsymbol{\theta}^{k,t}, k \in \mathcal{K}_t$, by using their local datasets $\mathcal{D}_k$, each composed of $D_k = |\mathcal{D}_k|$ samples. These local masks are then aggregated by the server in a communication-efficient way to estimate the optimal $\boldsymbol{\theta}$. At test time, at the server, the initial random network $f_{\boldsymbol{w}^{\text{init}}}$ is sparsified using the global probability mask $\boldsymbol{\theta}^{g,t}$, following the stochastic approach in Figure 1. In the following sections, we provide more details on each step of each round. We give the pseudocode for FedPM in Appendix A.

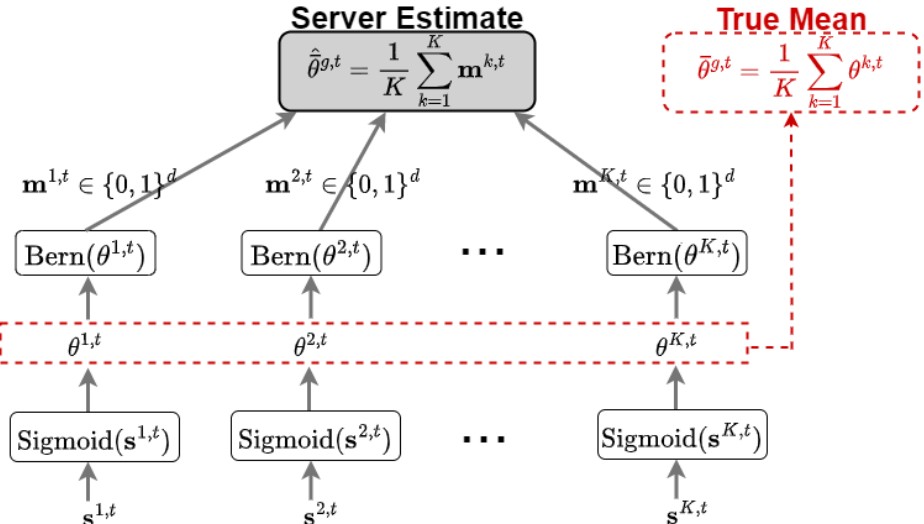

Figure 2: Communication-efficient estimation of the mean of the probability masks $\bar{\boldsymbol{\theta}}^{g,t}$. Each client communicates a stochastic binary mask $\boldsymbol{m}^{k,t}$ sampled from the local Bernoulli mask $\boldsymbol{\theta}^{k,t}$. We reduce the bitrate to less than 1 bit per parameter by using arithmetic coding to encode $\boldsymbol{m}^{k,t}$. When the frequency of 1's is far from 0.5 (which is usually the case with FedPM), the number of bits per parameter to communicate $\boldsymbol{m}^{k,t}$ is less than 1. See Figure 3 for more details.

#### 2.1.1 Local Training of Probability Masks

Upon receiving a global probability mask $\boldsymbol{\theta}^{g,t-1}$ from the server at the beginning of round $t$, the client $k$ performs local training and updates the mask via back-propagation. First, however, we have to guarantee that the updated probability mask satisfies $\boldsymbol{\theta}^{k,t} \in [0,1]^d$. While this can be achieved with a regularization term or clipping, these approaches may lead to a performance drop due to information loss. Therefore, similarly to the work of Zhou et al. (2019), we introduce another mask, called *score mask* $\boldsymbol{s} = (s_1, s_2, \ldots, s_d) \in \mathbb{R}^d$, that has unbounded support and that can be used to generate the probability masks through the one-to-one sigmoid function by setting $\boldsymbol{\theta} = \text{Sigmoid}(\boldsymbol{s})$. Then, the procedure for local training of the probability mask at round $t$ is as follows (here, the steps from Step 2 to 4 describe one local iteration, which is repeated a number $\tau$ of times as standard in FL (McMahan et al., 2017a)):

1. The server sends the global probability mask $\boldsymbol{\theta}^{g,t-1}$ to $K$ chosen clients, and the clients set $\boldsymbol{s}^{k,t} = \text{Sigmoid}^{-1}(\boldsymbol{\theta}^{g,t-1})$, where $\text{Sigmoid}^{-1}(\cdot)$ is the inverse of the sigmoid function.

2. Then, the clients generate a binary mask by first transforming back $\boldsymbol{\theta}^{k,t} = \text{Sigmoid}(\boldsymbol{s}^{k,t})$, and then sampling a binary mask $\boldsymbol{M}^{k,t}$ from $\boldsymbol{\theta}^{k,t}$ as shown in Figure 1: $\boldsymbol{m}^{k,t} \sim \text{Bern}(\boldsymbol{\theta}^{k,t})$.

3. The sampled binary mask then sparsifies the initial weight vector $\boldsymbol{w}^{\text{init}}$: $\dot{\boldsymbol{w}}^{k,t} = \boldsymbol{m}^{k,t} \odot \boldsymbol{w}^{\text{init}}$.

4. $\dot{\boldsymbol{w}}^{k,t}$ is then used for forward pass, and the loss $\mathcal{L}(f_{\dot{\boldsymbol{w}}^{k,t}}, \mathcal{D}_k)$ on the local task is backpropagated to update the score mask as $\boldsymbol{s}^{k,t} = \boldsymbol{s}^{k,t} - \eta \nabla \mathcal{L}(f_{\dot{\boldsymbol{w}}^{k,t}}, \mathcal{D}_k)$ ($\eta$ is the local learning rate).

All the local operations from Step 2 to Step 4 are differentiable, except for the Bernoulli sampling. To backpropagate the gradients through the sampling operation, we use the first-order gradient of the Bernoulli function, which is simply equal to the probability mask $\boldsymbol{\theta}^{k,t}$.

#### 2.1.2 Communication Strategy

Once the local training at round $t$ is completed, the server needs to distill the global probability mask $\boldsymbol{\theta}^{g,t}$, for example, by taking the empirical average of the local probability masks $\bar{\boldsymbol{\theta}}^{g,t} = \frac{1}{K} \sum_{k \in \mathcal{K}_t} \boldsymbol{\theta}^{k,t}$ from the clients. However, since we aim for communication efficiency, the clients do not send their local probability masks directly. Instead, they communicate a stochastic binary sample

$M^{k,t}$ from their probability masks sampled as $m^{k,t} \sim \text{Bern}(\theta^{k,t})$, and then the server estimates the global aggregate $\bar{\theta}^{g,t}$ as $\hat{\bar{\theta}}^{g,t} = \frac{1}{K} \sum_{k \in \mathcal{K}_t} m^{k,t}$. This distributed mean estimation problem with communication constraints is summarized in Figure 2. Our estimator $\hat{\bar{\theta}}^{g,t} = \frac{1}{K} \sum_{k \in \mathcal{K}_t} m^{k,t}$ is an unbiased estimate of the true aggregate, in that

$$
\begin{aligned}
\mathbb{E}_{M^{k,t} \sim \text{Bern}(\theta^{k,t}) \ \forall k \in \mathcal{K}_t}[\hat{\bar{\theta}}^{g,t}] &= \mathbb{E}_{M^{k,t} \sim \text{Bern}(\theta^{k,t}) \ \forall k \in \mathcal{K}_t} \left[ \frac{1}{K} \sum_{k \in \mathcal{K}_t} M^{k,t} \right] \\
&= \frac{1}{K} \sum_{k \in \mathcal{K}_t} \mathbb{E}_{M^{k,t} \sim \text{Bern}(\theta^{k,t})}[M^{k,t}] \\
&= \frac{1}{K} \sum_{k \in \mathcal{K}_t} \theta^{k,t} \\
&= \bar{\theta}^{g,t}.
\end{aligned}
$$

Moreover, the estimation error is upper bounded as (the proof is given in Appendix B)

$$
\mathbb{E}_{M^{k,t} \sim \text{Bern}(\theta^{k,t}) \ \forall k \in \mathcal{K}_t} \left[ ||\hat{\bar{\theta}}^{g,t} - \bar{\theta}^{g,t}||_2^2 \right] \leq \frac{d}{4K}. \tag{2}
$$

Since each client communicates a stochastic binary mask $M^{k,t}$, 1 bpp is the worst case bitrate for FedPM. We can further reduce the bitrate to less than 1 bit by using arithmetic coding (Rissanen & Langdon, 1979) or universal coding (Krichevsky & Trofimov, 1981; Barron et al., 1998) to encode $m^{k,t}$, and achieve the empirical entropy since $d$ is large. This gives us smaller bitrates whenever the frequency of 1's in $m^{k,t}$ is far from 0.5 – which is usually the case for our method (see Figure 3 and Appendix E.1 for results). We note that, with a deterministic mask training approach as in FedMask (Li et al., 2021), arithmetic coding of $m^{k,t}$s does not provide any further gain in bitrate, as we have empirically observed that the frequency of 1's is always around 0.5 (see Figure 3 and Appendix E.1) – here we apply arithmetic coding for FedMask to improve our baseline although it was not proposed in the original paper. Moreover, FedMask (Li et al., 2021) and HideNSeek (Vallapuram et al., 2022) do not enjoy the guarantees we have as their estimator (i) is not unbiased and (ii) does not have an upper bound on the estimation error due to hard thresholding (Li et al., 2021) and sign operations (Vallapuram et al., 2022). This is another benefit of our stochastic sampling approach.

## 2.2 FedPM with Bayesian Aggregation

Another important aspect that differentiates our work from existing masking methods such as FedMask (Li et al., 2021) and HideNSeek (Vallapuram et al., 2022) is the Bayesian aggregation strategy, which exploits the underlying *stochastic mask* to synthesize a global model, boosting the performance in heterogeneous scenarios, e.g., when local client data are not sampled from the same distribution. Given the probabilistic interpretation of the FedPM mask's values, at the server side we further model the probability mask $\theta^{g,t}$ with a Beta distribution $\text{Beta}(\alpha^{g,t}, \beta^{g,t})$, parameterized by the round-dependent parameters $\alpha^{g,t}$ and $\beta^{g,t}$, which are initialized to $\alpha^{g,0} = \beta^{g,0} = \lambda_0$. At the beginning of the training process, there is no prior knowledge indicating which network weight should be more important than the others, and so each entry in the probability mask is uniformly distributed in $[0, 1]$ – which is the *prior* distribution. Consequently, the clients' local binary masks $M^{k,t}$s are the *data* the server uses to update its belief on each weight score, and so the aggregation strategy corresponds now to a *posterior update*. Specifically, given the conjugate relation between the Beta-Bernoulli distributions, the new posteriors are still Beta distributions with parameters

$$
\alpha^{g,t} = \alpha^{g,t-1} + M^{\text{agg},t} \quad \text{and} \quad \beta^{g,t} = \beta^{g,t-1} + K \cdot \mathbf{1} - M^{\text{agg},t} \quad \forall t \geq 1, \tag{3}
$$

where $M^{\text{agg},t} = \sum_{k \in \mathcal{K}_t} M^{k,t}$, and $\mathbf{1}$ is the $d$-dimensional vector containing all ones. Then, the server broadcasts to the clients the mode of the Bernoulli distributions, as suggested by Ferreira et al. (2021), i.e.,

$$
\theta^{g,t} = \frac{\alpha^{g,t} - 1}{\alpha^{g,t} + \beta^{g,t} - 2}, \tag{4}
$$

where the division operation is applied element-wise. However, to obtain the best performance out of this method, the Beta parameters should be re-initialized to their original values $\boldsymbol{\lambda}_0$ with some regularity. Notice that if $\boldsymbol{\lambda}_0 = \mathbf{1}$, and if $\boldsymbol{\alpha}$ and $\boldsymbol{\beta}$ are re-initialized at the beginning of each round, the method is equivalent to the aggregation strategy detailed in Section 2.1.2.

## 2.3  Weight Distribution

As mentioned in Section 2.1, the fixed weight vector $\boldsymbol{w}^{\text{init}}$ is initialized by sampling from the distribution $P_w$ using the randomly generated SEED. We note that the choice of this distribution impacts two important aspects of FedPM: (i) the values of $\boldsymbol{w}^{\text{init}}$ highly influence the final accuracy achieved by the model, as they represent the building blocks to extract a subnetwork $f_{\hat{\boldsymbol{w}}}$ (see Figure 1), which should be rich enough to solve the learning task, and (ii) the size of the sample space of $P_w$ affects the number of bits needed to store the model during the *inference process* (this is different from the 1 bpp model storage when the model is not in use). Regarding (i), as also proposed in Ramanujan et al. (2020), we sample weights from a uniform distribution, whose domain is $\{-\sigma, +\sigma\}$, where $\sigma$ is the standard deviation of the Kaiming Normal distribution (He et al., 2015). In this way, we control the variance of the neurons' output to be $\sim 1$, which avoids the vanishing or the explosion of activation values. Previous experiments in (Zhou et al., 2019; Ramanujan et al., 2020) also demonstrate the superior performance achieved by binary weights distributions when compared to standard continuous counterparts, e.g., Gaussian. Regarding (ii), even if knowing the value of SEED is enough to perfectly reconstruct the vector $\boldsymbol{w}^{\text{init}}$, one would have to generate the entire vector at every inference step. Consequently, to achieve fast inference, the actual values of the weights need to be stored in the memory of the devices during the *inference process*. Fortunately, our initialization allows for efficient storage even during inference since (after reconstructing $\boldsymbol{w}^{\text{final}}$ using SEED and $\boldsymbol{m}^{\text{final}} \in \{0,1\}^d$) we only need to indicate whether the weight values in $\boldsymbol{w}^{\text{final}}$ are $-\sigma$, $0$, or $+\sigma$, with a ternary representation that can be efficiently deployed on hardware (Alemdar et al., 2017).

## 2.4  Privacy

Privacy is another challenge in FL as the model updates (in our case, $\boldsymbol{M}^{k,t}$s) may leak information about the client data. Differential privacy (DP) guarantees that the probability of an outcome of an algorithm that runs on client data does not change much by a single client's data (see Appendix C for the formal definition). This is typically ensured via injecting noise to a function of the client data at a particular step in the algorithm with some utility loss in the application. While there have been many DP strategies developed for FL and deep learning (Abadi et al., 2016; McMahan et al., 2017b; Agarwal et al., 2021; Andrew et al., 2021), these strategies typically suffer from severe performance degradation due to noise injection. To make DP practical, researchers have explored certain randomization mechanisms that amplify the privacy guarantee. When these mechanisms are part of the FL framework, such as sampling (data (Balle et al., 2018; Wang et al., 2019) or device (Balle et al., 2020; Girgis et al., 2021; Hasircioglu & Gunduz, 2022)) and shuffling (Erlingsson et al., 2019; Feldman et al., 2022), the amplification comes for free. This is helpful because the overall process can meet a stronger privacy guarantee without increasing the noise level. FedPM promises one such amplification due to the stochastic Bernoulli sampling step. In particular, Imola & Chaudhuri (2021) have shown that when a sample $\boldsymbol{M} \in \{0,1\}^d$ from an already privatized vector $\boldsymbol{\theta} \in [c, 1-c]^d$, where $0 < c < 0.5$, is released to a third party (instead of $\boldsymbol{\theta}$ itself), the privacy is amplified under some conditions. More precisely, when there is an $(\alpha, \epsilon)$-Rényi Differential Privacy mechanism (Mironov, 2017) that privatizes $\boldsymbol{\theta} \in [c, 1-c]^d$, releasing a sample from $\text{Bern}(\boldsymbol{\theta})$ yields an improved privacy budget (the smaller $\epsilon$, the better the privacy): $\epsilon_{amp} \leq \min\{\epsilon, d \cdot r_\alpha(c)\}$. Here, $r_\alpha(p)$ is the binary symmetric Rényi divergence function defined as $r_\alpha(p) = \frac{1}{\alpha-1} \log\left(p^\alpha(1-p)^{1-\alpha} + (1-p)^\alpha p^{1-\alpha}\right)$. Notice that FedPM already involves this Bernoulli sampling step in the communication protocol and in the forward pass $\boldsymbol{m}^{k,t} \sim \text{Bern}(\boldsymbol{\theta}^{k,t})$. However, the $d$ term in the upper bound limits the amplification for large model sizes. We believe it is worth exploring a tighter upper bound on $\epsilon_{amp}$ to enjoy privacy amplification in FedPM with practical models. Nonetheless, in Appendix C, we demonstrate the impact of this amplification on a distributed mean estimation problem, described in Figure 2, where the goal is to estimate the true mean of the probability masks $\bar{\boldsymbol{\theta}} = \frac{1}{K}\sum_{k=1}^K \boldsymbol{\theta}^k$ under communication and privacy constraints. We also provide a bias correction mechanism, specific to our scheme in Figure 4 in Appendix C, that mitigates the bias due to the DP mechanism and reduces the estimation error.

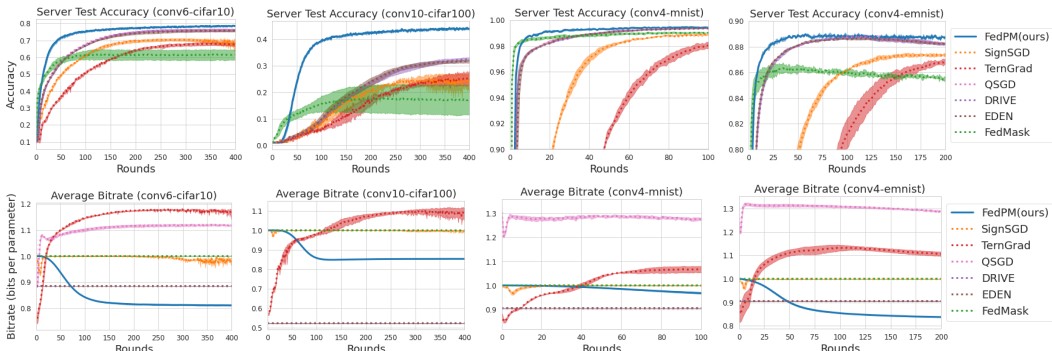

Figure 3: Accuracy and bitrate comparison of FedPM with baselines SignSGD (Bernstein et al., 2018), TernGrad (Wen et al., 2017), QSGD (Alistarh et al., 2017), DRIVE (Vargaftik et al., 2021), EDEN (Vargaftik et al., 2022), and FedMask (Li et al., 2021), all performing in the same bitrate regime, on CIFAR-10, CIFAR-100, MNIST, and EMNIST datasets. Note that the bitrate in QSGD is adjustable via the number of levels. In these plots, we pick a level number that gives bitrate slightly larger than 1. However, the accuracy is still lower than FedPM accuracy. It is possible to reduce the bitrate in QSGD and still have some reasonable accuracy; however, the performance in that regime (lower bitrate, lower accuracy) is not a meaningful comparison point for FedPM.

## 3 Experiments

In this section, we empirically show the performance of FedPM in terms of accuracy, bitrate, converge speed, and the final model size. We consider four datasets: CIFAR-10 with 10 classes, CIFAR-100 (Krizhevsky et al., 2009) with 100 classes, MNIST (Deng, 2012) with 10 classes, and EMNIST (Cohen et al., 2017) with 47 classes. For CIFAR-100, we use a 10-layer convolutional network (CNN) CONV-10; for CIFAR-10, a 6-layer CNN CONV-6; and for MNIST and EMNIST, a 4-layer CNN CONV-4. A detailed description of the architectures can be found in Appendix D. We first compare FedPM with SignSGD (Bernstein et al., 2018), TernGrad (Wen et al., 2017), QSGD (Alistarh et al., 2017), DRIVE (Vargaftik et al., 2021), EDEN (Vargaftik et al., 2022), and FedMask (Li et al., 2021) on IID data split and full client participation in Section 3.1. We then extend our experiments to non-IID data splits and partial participation in Section 3.2. Clients perform 3 local epochs in all experiments. We provide additional details on the experimental setup in Appendix D. We present results averaged over 3 runs.

### 3.1 IID Data Split and Full Participation ($K = N$)

In this section, we focus on IID data distribution and the case when all the clients participate in the training at each round. We set the number of clients to $N = K = 10$. We report the estimated bitrate for the arithmetic code that uses the empirical frequency of the symbols (for our method FedPM, this corresponds to the frequency of 1's in $m^{k,t}$) – which is equal to the empirical entropy for blocklength $d$ as large as the model size. In Figure 3, we compare the accuracy, bitrate, and convergence speed of FedPM with relevant baselines. As can be seen in the figure, FedPM converges to the highest accuracy on all four datasets. DRIVE, EDEN, and QSGD (they mostly overlap in the accuracy plots) seem to be the three baselines that perform the best after FedPM; however, their convergence speed is significantly lower than FedPM. In terms of convergence speed, FedMask is the fastest among the baselines – in fact, at the beginning of the training, FedMask is faster than FedPM as well. However, its final accuracy is lower than the others.

In terms of bitrate, SignSGD and FedMask consistently spend 1 bpp, which is the default number when a binary mask or sign mask is communicated. This means binary values (1's and 0's) are almost equally distributed in their masks, which prevents them from enjoying additional bitrate gains. Across all experiments, TernGrad has the highest bitrate. We would like to leave a note about the bitrate of QSGD. Unlike other baselines, including our work, QSGD can go down to very low bitrates by adjusting the number of levels in quantization. We have observed that in the extreme quantization case, QSGD underperforms FedPM. Then, we have decided to increase the number of quantization levels in QSGD to see if it improves the accuracy. However, as can be seen from the plots, even

with bitrate larger than 1, QSGD still underperforms FedPM. The only two baselines that challenge FedPM in terms of bitrate are DRIVE and EDEN. While FedPM has lower bitrates on CIFAR-10 and EMNIST; DRIVE and EDEN have better bitrates on CIFAR-100 and MNIST. However, the accuracy of DRIVE and EDEN on these datasets (specifically CIFAR-100) is significantly lower than that of FedPM, with slower convergence.

As for the final model size, FedPM needs only 0.8 bpp for the `CONV-6` model trained on CIFAR-10, 0.85 bpp for the `CONV-10` model trained on CIFAR-100, 0.96 bpp for the `CONV-4` model trained on MNIST, and 0.83 bpp for the `CONV-4` model trained on EMNIST. On the other hand, other baselines that train a dense model, namely SignSGD, TernGrad, QSGD, DRIVE, and EDEN, would need to represent each weight with their full precision value, i.e., 32 bpp. This implies that FedPM provides around $38.6\times$ improvement in the storage or the communication of the final model. Since FedMask also trains a sparse model, it enjoys a similar gain in the final model size requiring 1 bpp across all the models. Due to the stochastic masking procedure and uneven distribution of 1's and 0's in the binary masks, FedPM has up to 0.17 bpp improvement over the deterministic procedure in FedMask, which adds up to a large gain due to the huge model size.

We would like to highlight that while some of our baselines, such as FedMask and TernGrad, have a visibly high variance in accuracy, FedPM shows stable training behavior across all experiments.

## 3.2 Non-IID Data Split and Partial Participation ($K < N$)

This section considers more realistic scenarios, in which the local clients' datasets are generated from slightly different data distributions. We focus on CIFAR-10 with `CONV-6`, and we compare FedPM against (i) the most promising baselines, which, based on the results of Section 3.1, are DRIVE, EDEN, and QSGD, and (ii) FedMask, as it is the only *sparse* baseline. To choose the size of each dataset $|\mathcal{D}_n| = D_n$, for each client $n \in \{1, \ldots, N\}$, an integer $j_n$ is sampled uniformly from $\{10, 11, \ldots, 100\}$. Then, a coefficient $p_n = \frac{j_n}{\sum_n j_j}$ is computed, which represents the size of the local dataset $D_n$ as a fraction of the size of the full dataset, i.e., the training set of CIFAR-10. In this way, highly unbalanced datasets can be generated from the central one. Moreover, since the task is a classification problem, we impose a maximum number of different labels, or classes, $c_{\max}$, that one client can see. Consequently, clients need cooperation to learn the statistics of other classes' distributions, as the test dataset contains samples from all classes. In addition, partial participation is also considered, meaning that at each round, the server uniformly samples a fraction $\rho = \frac{K}{N}$ of the clients to participate in the training round. This is motivated in real-world scenarios by the scarcity of physical communication network resources, which may limit the availability of part of the clients during one round. The maximum number of classes per local dataset is set to $c_{\max} \in \{2, 4\}$, and the participation ratio is set to $\rho \in \{0.1, 0.5, 1\}$. For $\rho = 1$ and $\rho = 0.5$, the total number of clients is set to $N = 10$ (and so $K$ is equal to 10 and 5, respectively). For $\rho = 0.1$, we set $N = 50$ (and so $K = 5$), which is the worst scenario among all combinations, given the small amount of information the server can collect at the end of each round. When $\rho = 1$, for the FedPM algorithm, we keep the same aggregation strategy exposed in Section 2.1.2 and Figure 2; and we switch to the Bayesian aggregation method (see Section 2.2) when there is partial participation, i.e., when $\rho < 1$. Indeed, applying the Bayesian aggregation method is revealed to be crucial for achieving good accuracy when $\rho < 1$ and data are non-IID, obtaining a large gain with respect to the simpler version in Section 2.1.2, which resets the Beta priors at each round (or takes the average of the samples, as explained in Section 2.2). We adopt a simple heuristic schedule to reset the priors: Reset every 3 rounds when $\rho = 0.5$, and every 10 rounds when $\rho = 0.1$. As expected, the smaller the ratio $\rho$, the larger the number of rounds we should wait before resetting the priors in order to collect more information from a much more diverse pool of clients.

Table 1 reports the results when $c_{\max} = 4$ and $c_{\max} = 2$. As we can see, FedPM outperforms all the baselines in every configuration, as the Bayesian aggregation allows the central node to collect more data before resetting the priors, which is important when clients' data distributions are heterogeneous. This strategy can be seen as the FedPM counterpart of decreasing the learning rate (which we applied in the other *dense* compression-based baselines, like DRIVE, EDEN, and QSGD). It is seen from Table 1 that FedMask (Li et al., 2021) is struggling in the non-IID case, as applying a hard threshold on the scores to binarize the mask does not provide a proper way to implement multiple-rounds aggregation, emphasizing the benefit of the stochastic process in FedPM. It is interesting to notice that, especially when $c_{\max} = 4$, the lower the value of $\rho$, the larger the gap between FedPM and the

Table 1: Average final accuracy $\pm\sigma$ in non-IID data split with $c_{\max} = 4$ and $c_{\max} = 2$, and partial participation with ratios $\rho = \{0.1, 0.5, 1\}$, for FedPM, FedMask, and the strongest baselines in the IID experiments: EDEN, DRIVE, and QSGD. The training duration was set to $t_{\max} = 200$ rounds.

|  | Algorithm | $\rho = 1$ | $\rho = 0.5$ | $\rho = 0.1$ |
|---|---|---|---|---|
| $c_{\max} = 4$ | DRIVE (Vargaftik et al., 2021) | $0.739 \pm 0.005$ | $0.632 \pm 0.010$ | $0.405 \pm 0.018$ |
|  | EDEN (Vargaftik et al., 2022) | $0.717 \pm 0.006$ | $0.665 \pm 0.012$ | $0.360 \pm 0.016$ |
|  | QSGD (Alistarh et al., 2017) | $0.709 \pm 0.006$ | $0.644 \pm 0.014$ | $0.399 \pm 0.020$ |
|  | FedMask (Li et al., 2021) | $0.531 \pm 0.044$ | $0.435 \pm 0.057$ | $0.362 \pm 0.024$ |
|  | FedPM (Ours) | $\mathbf{0.748 \pm 0.003}$ | $\mathbf{0.720 \pm 0.007}$ | $\mathbf{0.496 \pm 0.007}$ |
| $c_{\max} = 2$ | DRIVE (Vargaftik et al., 2021) | $0.434 \pm 0.025$ | $0.376 \pm 0.014$ | $0.221 \pm 0.003$ |
|  | EDEN (Vargaftik et al., 2022) | $0.535 \pm 0.050$ | $0.461 \pm 0.016$ | $0.219 \pm 0.005$ |
|  | QSGD (Alistarh et al., 2017) | $0.476 \pm 0.033$ | $0.464 \pm 0.002$ | $0.243 \pm 0.014$ |
|  | FedMask (Li et al., 2021) | $0.420 \pm 0.028$ | $0.387 \pm 0.062$ | $0.197 \pm 0.030$ |
|  | FedPM (Ours) | $\mathbf{0.643 \pm 0.016}$ | $\mathbf{0.556 \pm 0.031}$ | $\mathbf{0.277 \pm 0.003}$ |

baselines, corroborating the fact that the Bayesian strategy can better deal with partial participation. Analysis of the communication bitrate is provided in Appendix. E.1.

## 4   Conclusion

In this work, we introduced Federated Probabilistic Mask Training (FedPM) – a communication-efficient FL strategy. FedPM relies on the idea of finding a sparse network in a randomly initialized dense network, which is then sparsified by a collaboratively trained *stochastic* binary mask. In addition to reducing the communication cost to less than 1 bit per parameter (bpp), FedPM also reaches higher accuracy with faster convergence than the relevant baselines, and can potentially amplify privacy while additionally outputting a compressed final model with a size less than 1 bpp. Throughout the manuscript, we highlighted the advantages of having a stochastic mask training approach rather than a deterministic one in terms of accuracy, bitrate, and privacy. For instance, the proposed Bayesian aggregation strategy boosts the performance of FedPM in the non-IID data split and/or the partial client participation case by using the prior knowledge from previous rounds during aggregation rather than hard-replacing the previous global probability mask with the new one.

## 5   Acknowledgement

The authors would like to thank Zachary Charles, Mahdi Haghifam, Peter Kairouz, and Nicole Mitchell for inspiring discussions. This work was supported in part by a Sony Stanford Graduate Fellowship and a Meta research award.

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

# A   FedPM Algorithm

We provide the pseudocode for FedPM in Algorithms 1 and 2. In Algorithm 2, the prior resetting scheduling policy is controlled by the procedure *ResPrior(t)*, which may depend on quantities other than the round number $t$, such as loss.

---

**Algorithm 1** FedPM.

---

**Hyperparameters:** learning rate $\eta$, minibatch size $B$, number of local iterations $\tau$.
**Inputs:** local datasets $\mathcal{D}_i$, $i = 1, \ldots, N$
**Output:** random seed SEED and binary mask parameters $\boldsymbol{m}^{k,T}$

At the server, initialize a random network with weight vector $\boldsymbol{w}^{\text{init}} \in \mathbb{R}^d$ using a random seed SEED, and broadcast it to the clients.
At the server, initialize the random score vector $\boldsymbol{s}^{g,0} \in \mathbb{R}^d$, and compute $\boldsymbol{\theta}^{g,0} \leftarrow \text{Sigmoid}(\boldsymbol{s}^{g,0})$.
At the server, initialize Beta priors $\boldsymbol{\alpha}^{g,0} = \boldsymbol{\beta}^{g,0} = \boldsymbol{\lambda}_0$.
**for** $t = 1, \ldots, T$ **do**
    Sample a subset $\mathcal{K}_t \subset \{1, \ldots, N\}$ of $|\mathcal{K}_t| = K$ clients without replacement.
    **On Client Nodes:**
    **for** $k \in \mathcal{K}_t$ **do**
        Receive $\boldsymbol{\theta}^{g,t-1}$ from the server and set $\boldsymbol{s}^{k,t} = \text{Sigmoid}^{-1}(\boldsymbol{\theta}^{g,t-1})$.
        **for** $l = 1, \ldots, \tau$ **do**
            $\boldsymbol{\theta}^{k,t} \leftarrow \text{Sigmoid}(\boldsymbol{s}^{k,t})$
            Sample binary mask $\boldsymbol{m}^{k,t} \sim \text{Bern}(\boldsymbol{\theta}^{k,t})$.
            $\dot{\boldsymbol{w}}^{k,t} \leftarrow \boldsymbol{m}^{k,t} \odot \boldsymbol{w}^{\text{init}}$
            $\text{grad}_{\boldsymbol{s}^{k,t}} \leftarrow \frac{1}{B} \sum_{b=1}^{B} \nabla \ell(\dot{\boldsymbol{w}}^{k,t}; \mathcal{B}_j^k)$; $\{\mathcal{B}_j^k\}_{j=1}^{B}$ is uniformly chosen from $\mathcal{D}_k$
            $\boldsymbol{s}^{k,t} \leftarrow \boldsymbol{s}^{k,t} - \eta \cdot \text{grad}_{\boldsymbol{s}^{k,t}}$
        **end for**
        $\boldsymbol{\theta}^{k,t} \leftarrow \text{Sigmoid}(\boldsymbol{s}^{k,t})$
        Sample a binary mask $\boldsymbol{m}^{k,t} \sim \text{Bern}(\boldsymbol{\theta}^{k,t})$.
        Send the arithmetic coded binary mask $\boldsymbol{m}^{k,t}$ to the server.
    **end for**

    **On the Server Node:**
    Receive $\boldsymbol{m}^{k,t}$'s from $K$ client nodes.
    $\boldsymbol{\theta}^{g,t} = \text{BayesAgg}(\{\boldsymbol{m}^{k,t}\}_{k \in \mathcal{K}_t}, t)$     // See Algorithm 2.
    Broadcast $\boldsymbol{\theta}^{g,t}$ to all client nodes.
**end for**
Sample the final binary mask $\boldsymbol{m}^{\text{final}} \sim \text{Bern}(\boldsymbol{\theta}^{g,T})$.
Generate the final model: $\dot{\boldsymbol{w}}^{\text{final}} \leftarrow \boldsymbol{m}^{\text{final}} \odot \boldsymbol{w}^{\text{init}}$.

---

---

**Algorithm 2** BayesAgg.

---

**Inputs:** clients' updates $\{\boldsymbol{m}^{k,t}\}_{k \in \mathcal{K}_t}$, and round number $t$
**Output:** global probability mask $\boldsymbol{\theta}^{g,t}$

**if** ResPriors($t$) **then**
    $\boldsymbol{\alpha}^{g,t-1} = \boldsymbol{\beta}^{g,t-1} = \boldsymbol{\lambda}_0$
**end if**
Compute $\boldsymbol{m}^{\text{agg},t} = \sum_{k \in \mathcal{K}_t} \boldsymbol{m}^{k,t}$.
$\boldsymbol{\alpha}^{g,t} = \boldsymbol{\alpha}^{g,t-1} + \boldsymbol{m}^{\text{agg},t}$
$\boldsymbol{\beta}^{g,t} = \boldsymbol{\beta}^{g,t-1} + K \cdot \mathbf{1} - \boldsymbol{m}^{\text{agg},t}$
$\boldsymbol{\theta}^{g,t} = \frac{\boldsymbol{\alpha}^{g,t}-1}{\boldsymbol{\alpha}^{g,t}+\boldsymbol{\beta}^{g,t}-2}$
Return $\boldsymbol{\theta}^{g,t}$

---

# B   Proof of the Upper Bound on the Estimation Error

We now provide proof of the upper bound on the estimation error in Eq. 2. Recall that our true mean is $\bar{\boldsymbol{\theta}}^{g,t} = \frac{1}{K}\sum_{k\in\mathcal{K}_t}\boldsymbol{\theta}^{k,t}$, whereas our estimate is $\bar{\boldsymbol{\theta}}^{g,t} = \frac{1}{K}\sum_{k\in\mathcal{K}_t}\boldsymbol{m}^{k,t}$, where $\boldsymbol{m}^{k,t} \sim \text{Bern}(\boldsymbol{\theta}^{k,t})$. Then we can compute the error as

$$\mathbb{E}_{\boldsymbol{M}^{k,t}\sim\text{Bern}(\boldsymbol{\theta}^{k,t})\ \forall k\in\mathcal{K}_t}\left[||\hat{\bar{\boldsymbol{\theta}}}^{g,t} - \bar{\boldsymbol{\theta}}^{g,t}||_2^2\right] = \sum_{i=1}^{d}\mathbb{E}_{M_i^{k,t}\sim\text{Bern}(\theta_i^{k,t})\ \forall k\in\mathcal{K}_t}\left[\left(\hat{\bar{\theta}}_i^{g,t} - \bar{\theta}_i^{g,t}\right)^2\right] \tag{5}$$

$$= \sum_{i=1}^{d}\mathbb{E}_{M_i^{k,t}\sim\text{Bern}(\theta_i^{k,t})\ \forall k\in\mathcal{K}_t}\left[\left(\frac{1}{K}\sum_{k\in\mathcal{K}_t}(M_i^{k,t} - \theta_i^{k,t})\right)^2\right] \tag{6}$$

$$= \frac{1}{K^2}\sum_{i=1}^{d}\mathbb{E}_{M_i^{k,t}\sim\text{Bern}(\theta_i^{k,t})\ \forall k\in\mathcal{K}_t}\left[\left(\sum_{k\in\mathcal{K}_t}(M_i^{k,t} - \theta_i^{k,t})\right)^2\right] \tag{7}$$

$$= \frac{1}{K^2}\sum_{i=1}^{d}\mathbb{E}_{M_i^{k,t}\sim\text{Bern}(\theta_i^{k,t})\ \forall k\in\mathcal{K}_t}\left[\sum_{k\in\mathcal{K}_t}\left(M_i^{k,t} - \theta_i^{k,t}\right)^2\right] \tag{8}$$

$$= \frac{1}{K^2}\sum_{i=1}^{d}\sum_{k\in\mathcal{K}_t}\mathbb{E}_{M_i^{k,t}\sim\text{Bern}(\theta_i^{k,t})}\left[(M_i^{k,t} - \theta_i^{k,t})^2\right] \tag{9}$$

$$= \frac{1}{K^2}\sum_{i=1}^{d}\sum_{k\in\mathcal{K}_t}\left(\mathbb{E}_{M_i^{k,t}\sim\text{Bern}(\theta_i^{k,t})}[(M_i^{k,t})^2] - (\theta_i^{k,t})^2\right) \tag{10}$$

$$= \frac{1}{K^2}\sum_{i=1}^{d}\sum_{k\in\mathcal{K}_t}\left(\theta_i^{k,t} - (\theta_i^{k,t})^2\right) \tag{11}$$

$$\leq \frac{d}{4K}. \tag{12}$$

From (5) to (6), we use the definition of $\hat{\bar{\theta}}_i^{g,t} = \frac{1}{K}\sum_{k=1}^{K}m_i^{k,t}$ and $\bar{\theta}_i^{g,t} = \frac{1}{K}\sum_{k=1}^{K}\theta_i^{k,t}$. From (7) to (8), we use the fact that $\mathbb{E}_{M_i^{k,t}\sim\text{Bern}(\theta_i^{k,t})\ \forall k\in\mathcal{K}_t}[M_i^{k,t} - \theta_i^{k,t}] = 0$; and $M_i^{k,t} - \theta_i^{k,t}$ and $M_i^{l,t} - \theta_i^{l,t}$ are independent for $l \neq k \in [K]$. Finally, the inequality in (8) follows from $\theta_i^{k,t} \in [0,1]$ for all $k \in [K]$.

# C   Privacy Amplification and Bias Correction

We first revisit the definitions of differential privacy (Dwork et al., 2006), Rényi divergence, and Rényi differential privacy (Mironov, 2017).

**Definition 1.** *[Adjacent Datasets] Two datasets $D, D' \in \mathcal{D}$ are called adjacent if they differ in at most one data sample.*

**Definition 2.** *[$(\epsilon, \delta)$-DP] A randomized mechanism $f : \mathcal{D} \to \mathcal{R}$ offers $(\epsilon, \delta)$-differential privacy if for any adjacent $D, D' \in \mathcal{D}$ and $\mathcal{S} \subset \mathcal{R}$*

$$Pr[f(D) \in \mathcal{S}] \leq e^{\epsilon}Pr[f(D' \in \mathcal{S})] + \delta.$$

**Definition 3.** *[Rényi Divergence] For two probability distributions $P$ and $Q$ defined over $\mathcal{R}$, the Rényi divergence of order $\alpha > 1$ is*

$$D_\alpha(P||Q) = \frac{1}{\alpha - 1}\log\mathbb{E}_{x\sim Q}\left(\frac{P(x)}{Q(x)}\right)^\alpha.$$

**Definition 4.** *[$(\alpha, \epsilon)$-RDP] A randomized mechanism $f : \mathcal{D} \to \mathcal{R}$ offers $\epsilon$-Rényi differential privacy of order $\alpha$ (or in short $(\alpha, \epsilon)$-RDP) if for any adjacent $D, D' \in \mathcal{D}$, it holds that*

$$D_\alpha(f(D)||f(D')) \leq \epsilon.$$

Now, recall our discussion in Section 2.4 that we have an $(\alpha, \epsilon)$-RDP algorithm $f$ that outputs privatized $\boldsymbol{\theta}^k \in [c, 1-c]^d$ using local client data $\mathcal{D}_k$. As summarized in Figure 4, we are interested in what happens when instead of releasing $\boldsymbol{\theta}^k = f(\mathcal{D}_k)$, the client $k$ releases a Bernoulli sample from it: $\boldsymbol{m}^k \in \{0,1\}^d \sim \text{Bern}(\boldsymbol{\theta}^k)$. We already explained the advantages in terms of communication bitrate, estimation error, unbiasedness throughout the manuscript; however, this approach also amplifies the privacy guarantees, meaning that it makes the overall privacy budget smaller $\epsilon_{amp} \leq \epsilon$. Quantitatively, Imola & Chaudhuri (2021) showed that after the Bernoulli sampling, the privacy budget of the overall process is

$$\epsilon_{amp} \leq \min \{\epsilon, dr_\alpha(c)\},$$

where $r_\alpha(\cdot)$ is the Rényi divergence of the binary symmetric function. More precisely, consider $P, Q$ random variables with support on $\{x_1, x_2\} \subset \Theta$ and let $p = \Pr[P = x_1], 1 - p = \Pr(Q = x_1)$. Then the Rényi divergence is defined as

$$r_\alpha(p) = R_\alpha(P, Q) = \frac{1}{\alpha - 1} \log \left( p^\alpha (1-p)^{1-\alpha} + (1-p)^\alpha p^{1-\alpha} \right).$$

This implies that the stochastic Bernoulli sampling step of FedPM improves the privacy guarantee without changing the privacy mechanism – e.g. without increasing the injected noise level. We demonstrate this through a distributed mean estimation problem given in Figure4. We consider a simple differential privacy (DP) strategy, where the probability masks $\boldsymbol{\theta}^k \in [c, 1-c]^d$ are a function of client data $\mathcal{D}_k$; and are first corrupted by Gaussian noise, and then clipped to the range $[c, 1-c]^d$. Our goal is, as before, to estimate the true mean $\bar{\boldsymbol{\theta}} = \frac{1}{K} \sum_{k \in \mathcal{K}_t} \boldsymbol{\theta}^k$ by averaging the sampled binary masks, i.e., $\hat{\bar{\boldsymbol{\theta}}} = \frac{1}{K} \sum_{k \in \mathcal{K}_t} \boldsymbol{m}^k$. Differently from our previous experiments, we have privacy constraints now, meaning that we want to guarantee $(\epsilon, \delta)$-DP by injecting a Gaussian noise with variance $\sigma^2 = \frac{2 \ln (1.25/\delta) \Delta_2^2}{\epsilon^2}$ with a small $\epsilon$, where $\delta \approx \frac{1}{N^2}$ and $\Delta_2$ is the $\ell_2$-sensitivity of the probability masks (in our case $\Delta_2 = (1 - 2c)\sqrt{d}$). We transfer the above amplification results in RDP to DP using the well-known relation:

**Remark C.1.** *Mironov (2017) showed that if $f$ is an $(\alpha, \epsilon)$-RDP mechanism, it also satisfies $(\epsilon + \frac{\log 1/\delta}{\alpha - 1}, \delta)$-DP for any $0 < \delta < 1$.*

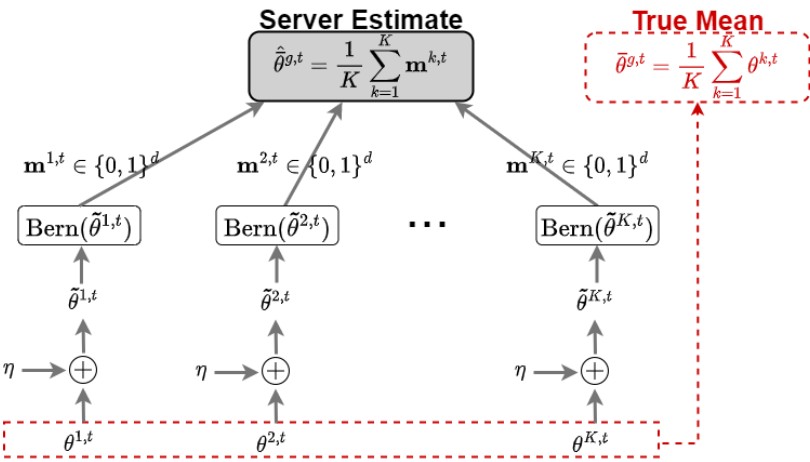

Figure 4: Distributed mean estimation scheme in FedPM, modified for differential privacy.

Since clipping after the noise addition step would lead to bias in the estimated mean, we work out a bias correction mechanism. We denote with $\theta$ one general parameter at client $k$ for one parameter, with $\tilde{\theta}$ its noisy version, and with $\hat{\theta} = \text{clip}(\tilde{\theta})$ its clipped version. Specifically, if $\tilde{\theta} = \theta + \eta$ is the

noisy version of the parameter, where $\eta \sim \mathcal{N}(0, \sigma^2)$, then

$$\text{clip}(\tilde{\theta}) = \begin{cases} \tilde{\theta}, & \text{if} \quad c \leq \theta + \eta \leq 1 - c \\ 1 - c, & \text{if} \quad \theta + \eta > 1 - c \\ c, & \text{if} \quad \theta + \eta < c. \end{cases} \tag{13}$$

We now compute $\mathbb{E}\left[\hat{M}\right]$, where $\hat{M} \sim \text{Bern}(\hat{\theta})$, to analyze the bias $\mathbb{E}\left[\hat{M}\right] - \mathbb{E}[M] = \mathbb{E}\left[\hat{M}\right] - \theta$, where $M \sim \text{Bern}(\theta)$. First of all, notice that

$$\mathbb{E}\left[\hat{M}\right] = \int_0^1 \mathbb{E}\left[\hat{M}|\hat{\theta} = \rho\right] f(\rho)d\rho = \int_0^1 \rho f(\rho)d\rho = \mathbb{E}[\hat{\theta}].$$

And we now compute the mean of the clipped parameter

$$\begin{aligned} \mathbb{E}\left[\hat{\theta}\right] &= \int_0^1 \rho f(\rho)d\rho \\ &= \int_{-\infty}^{+\infty} \text{clip}(\theta + \eta)f(\eta)d\eta \\ &= \int_{-\infty}^{c-\theta} c \cdot f(\eta)d\eta + \int_{c-\theta}^{1-c-\theta} (\theta + \eta) \cdot f(\eta)d\eta + \int_{1-c-\theta}^{+\infty} (1-c) \cdot f(\eta)d\eta \\ &= c \cdot \Phi_\sigma(c-\theta) + \theta \int_{c-\theta}^{1-c-\theta} f(\eta)d\eta + \int_{c-\theta}^{1-c-\theta} \eta f(\eta)d\eta + (1-c)\left(1 - \Phi_\sigma\left(1 - c - \theta\right)\right) \\ &= c \cdot \Phi_\sigma(c-\theta) + \theta\left[\Phi_\sigma(1-c-\theta) - \Phi_\sigma(c-\theta)\right] + \frac{-\sigma}{\sqrt{2\pi}}\left[e^{\frac{-(1-c-\theta)^2}{2\sigma^2}} - e^{\frac{-(c-\theta)^2}{2\sigma^2}}\right] + \\ & \quad + (1-c)\left(1 - \Phi_\sigma\left(1 - c - \theta\right)\right) \\ &= 1 - c + [\theta - 1 + c]\Phi_\sigma(1 - c - \theta) + [c - \theta]\Phi_\sigma(c-\theta) + \frac{-\sigma e^{\frac{-(c-\theta)^2}{2\sigma^2}}}{\sqrt{2\pi}}\left[e^{-2(c-\theta)-1} - 1\right], \end{aligned}$$

where $\Phi_\sigma(\cdot)$ is the cumulative distribution function of a Gaussian random variable with standard deviation $\sigma$, and zero mean. We use this relation to correct the bias in $\hat{\bar{\theta}}$.

We conduct our experiments with $N = 100$ clients, each having independent probability masks with dimension $d = 5$ and range $[0.2, 0.8]$, i.e., $\boldsymbol{\theta} \in [0.2, 0.8]^5$. Figure 5 shows the estimation error $||\hat{\bar{\boldsymbol{\theta}}}^{g,t} - \bar{\boldsymbol{\theta}}^{g,t}||_2^2$ under no noise injection case (i.e. no DP) with the black line. Recall that we want to reach a smaller estimation error and smaller $\epsilon$ (i.e., a stronger privacy guarantee). The red curve corresponds to the $\epsilon$ vs. estimation error behavior if Bernoulli sampling did not amplify the privacy. The blue curve shows the amplified $\epsilon$ (i.e. $\epsilon_{amp} \leq \epsilon$) vs. estimation error behavior, and it overlaps with the red curve for $\epsilon$ values smaller than $d \cdot r_\alpha(c) = 8.96$, where there is no privacy amplification, i.e., $\epsilon_{amp} = \epsilon$. However, notice that the blue line never reaches $\epsilon$'s higher than this value due to amplification, while enjoying smaller estimation errors that the red curve can only achieve with very large $\epsilon$. This shows the promise of FedPM in having a better privacy-accuracy performance than most baselines that do not have amplification. Finally, the green curve shows that bias correction improves this performance further even with $\epsilon < d \cdot r_\alpha(c) = 8.96$ by achieving lower estimation errors with the same $\epsilon$.

## D    Additional Experimental Details

In Table 2, we provide the architectures for all the models used in our experiments. Clients performed 3 local epochs with a batch size of 128 and a local learning rate of 0.1 in all the experiments. Notice that there is no server learning rate in FedPM; instead, we tune the prior resetting schedule in Bayesian aggregation for the experiments in Section 3.2. We conducted our experiments on NVIDIA Titan X GPUs on an internal cluster server, using 1 GPU per one run.

In the non-IID and partial participation experiments in Section 3.2, to distill the final model, we may apply both stochastic sampling, as during training, or a hard-threshold method, similar to the one

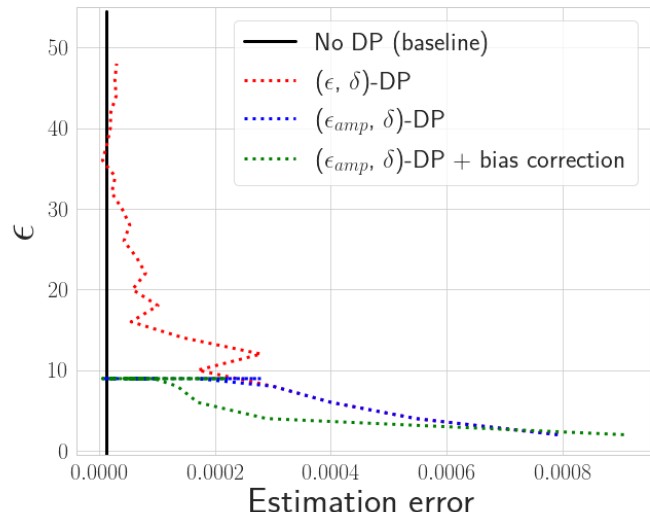

Figure 5: The effect of privacy amplification and bias correction in the privacy budget ($\epsilon$) vs. estimation error behavior. Comparing red and blue curves, we see that we can reach small estimation errors without increasing $\epsilon$ thanks to the amplification (see the vertical blue line at low estimation error.). While the red curve and blue curve overlap for $\epsilon < d \cdot r_\alpha(c) = 8.96$, in that regime, we benefit from our bias correction strategy to reach a lower error.

Table 2: Architectures for CONV-4, CONV-6, and CONV-10 models used in the experiments.

| Model | CONV-4 | CONV-6 | CONV-10 |
|---|---|---|---|
| Convolutional Layers | 64, 64, pool 128, 128, pool | 64, 64, pool 128, 128, pool 256, 256, pool | 64, 64, pool 128, 128, pool 256, 256, pool 512, 512, pool 1024, 1024, pool |
| Fully-Connected Layers | 256, 256, 10 | 256, 256, 10 | 256, 256, 100 |

adopted in FedMask (Li et al., 2021). In the latter, a binary mask coefficient $m_i$ is set to 1 if $\theta_i > \alpha_{\text{ths}}$, and 0 otherwise. For all experiments but one, when $\alpha_{\text{ths}} \in [0.4, 0.6]$, the thresholding test accuracy is always higher than the sampling method, and so we use the threshold method. However, in the extreme case $c_{\max} = 2$ and $\rho = 0.1$, the optimal values for $\alpha_{\max}$ were in $[0.2, 0.4]$ and $[0.6, 0.8]$ in all experiments, probably due to the high randomness given by the highly heterogeneous scenario. Consequently, for the last experiment, we just adopt the stochastic sampling strategy to evaluate the model, as further optimizing the $\alpha_{\text{ths}}$ means adapting to the test dataset, which may corrupt the ability of the model to generalize.

# E   Additional Experimental Results

## E.1   Bitrate Considerations on non-iid Data

We now report the communication bitrate considerations on the non-IID data split experiments described in Section 3.2. Table 3 reports the average bitrate needed by different algorithms over the whole training process when $c_{\max} = 4$ and $c_{\max} = 2$. By simply multiplying the obtained average bitrate by the total number of rounds $t_{\max} = 200$, we obtain the total number of bits one element

in the global probability mask needs to converge to its final value, indicating the total amount of information communicated during the training process.

We first observe that both DRIVE and EDEN consume almost the same amount of bits no matter the system configuration and round number (very small variance), and it is instead model dependent (see Figure 3). On the contrary, FedPM and QSGD report higher bitrate variability, as it depends on both the training phase and system setting. As already observed in Section 3.1, FedMask balances almost uniformly the binary updates, leading to a bitrate that is basically fixed to 1. For both $c_{max} = 4$ and $c_{max} = 2$, FedPM yields the smallest bitrate when $\rho = 1$, whereas for the other scenarios, EDEN and DRIVE are slightly more efficient. We argue that this is motivated by the fact that, as the learning task becomes harder due to the high system heterogeneity, all the models struggle to converge to good and stable solutions, which means that FedPM is still uncertain about the *weights' importance probabilities* $\boldsymbol{\theta}$, setting many of them close to 0.5. However, we think that this may be a useful feature of FedPM to quantify its internal uncertainty, which we will further analyze.

To conclude the analysis, we also report the FedPM bpp for the final model, which is an indication of the average number of bits needed per one parameter of the model. In the case of $c_{max} = 4$, the final model sizes are 0.79 bpp, 0.834 bpp, and 0.99 bpp, when $\rho = \{0.1, 0.5, 1\}$, respectively. When $c_{max} = 2$, the final model sizes are 0.8 bpp, 0.817 bpp, and 0.992 bpp. Consequently, at the end of the training process, FedPM remains the most efficient option, as already observed in Section 3.1.

Table 3: Average bitrate $\pm \sigma$ over the whole training process in non-IID data split with $c_{max} = 4$ and $c_{max} = 2$, and partial participation with ratios $\rho = \{0.1, 0.5, 1\}$, for FedPM, FedMask, and the strongest baselines in the IID experiments: EDEN, DRIVE, and QSGD. The training duration was set to $t_{max} = 200$ rounds.

|  | Algorithm | $\rho = 1$ | $\rho = 0.5$ | $\rho = 0.1$ |
|---|---|---|---|---|
| $c_{max} = 4$ | DRIVE (Vargaftik et al., 2021) | $0.885 \pm 9 \cdot 10^{-5}$ | $\mathbf{0.885 \pm 1 \cdot 10^{-4}}$ | $\mathbf{0.885 \pm 1 \cdot 10^{-4}}$ |
|  | EDEN (Vargaftik et al., 2022) | $0.885 \pm 1 \cdot 10^{-4}$ | $\mathbf{0.885 \pm 1 \cdot 10^{-4}}$ | $\mathbf{0.885 \pm 1 \cdot 10^{-4}}$ |
|  | QSGD (Alistarh et al., 2017) | $0.982 \pm 0.027$ | $0.923 \pm 0.029$ | $0.91 \pm 0.05$ |
|  | FedMask (Li et al., 2021) | $1 \pm 3 \cdot 10^{-6}$ | $1 \pm 8 \cdot 10^{-8}$ | $1 \pm 6 \cdot 10^{-7}$ |
|  | FedPM (Ours) | $\mathbf{0.863 \pm 0.077}$ | $0.912 \pm 0.056$ | $0.996 \pm 0.003$ |
| $c_{max} = 2$ | DRIVE (Vargaftik et al., 2021) | $0.885 \pm 7 \cdot 10^{-5}$ | $\mathbf{0.885 \pm 2 \cdot 10^{-4}}$ | $\mathbf{0.885 \pm 2 \cdot 10^{-4}}$ |
|  | EDEN (Vargaftik et al., 2022) | $0.885 \pm 1 \cdot 10^{-4}$ | $\mathbf{0.885 \pm 7 \cdot 10^{-5}}$ | $\mathbf{0.885 \pm 7 \cdot 10^{-5}}$ |
|  | QSGD (Alistarh et al., 2017) | $1.230 \pm 0.043$ | $1.234 \pm 0.038$ | $1.082 \pm 0.01$ |
|  | FedMask (Li et al., 2021) | $1 \pm 2 \cdot 10^{-6}$ | $1 \pm 2 \cdot 10^{-6}$ | $1 \pm 2 \cdot 10^{-7}$ |
|  | FedPM (Ours) | $\mathbf{0.868 \pm 0.076}$ | $0.904 \pm 0.063$ | $0.997 \pm 0.01$ |

