# OpenReview forum: "Efficient Federated Random Subnetwork Training"
_NeurIPS.cc/2022/Workshop/Federated_Learning — FL-NeurIPS 2022 Oral_

### Official Review · Reviewer_GSHz · 2022-10-17
**Solid workshop contribution**

I enjoyed reading this paper. The algorithm and analysis is comprehensive, the results convincing.
I would love to see experiments with more interesting/up-to-date model architectures. Are large models necessary to achieve the observed compression or can smaller, purely convolutional models such as ResNets benefit similarly?
One missing component to the experimental analysis is an ablation to the "Bayesian Aggregation" strategy: What happens without applying it? What happens without resetting ever-so-often and why do we observe this behavior?
I'm curious around your DP discussion: Why did you not provide experiments for FedPM with DP on a multi-round FL problem? Also why is the red-curve in Figure 5 not monotonically decreasing?

At the end of page 17, it should read e.g. '... with \tilde{\theta}} the noisy version and with ...'
Good luck with the full submission at ICLR

---

### Official Review · Reviewer_L3xR · 2022-10-17
**Paper is a bit hard to follow, lack of ablations and further experiments**

This paper considers using a Bernoulli probabilistic mask for training in federated learning. The general idea is straightforward. But I do have a few concerns:

1. This paper is not well-written and hard to follow. The general idea mentioned above is clear, but the design and further details are very hard to follow. The authors should consider simplifying the representation and using fewer symbols in a clearer and more readable way.

2. More ablations on different masking strategies should be considered. Different initialization strategy is also an interesting topic to explore.

---

### Official Review · Reviewer_BUsG · 2022-10-19
**The experimental work with good idea of mask training**

The paper proposes an interesting technique for communication efficiency using probabilistic mask training. The idea looks promising. However, the paper does not have any strong theory. At the same time the experimental comparison is wide and well-designed. It contains both iid and non-iid settings, also it has partial and full participation regimes.

I suggest to provide some convergence guarantees for the proposed method.

Overall, I believe this paper is good enough to be presented.

---

### Decision · Program_Chairs · 2022-10-20

Accept (Oral)